# SUBJECT CLUSTERING BY AN IMPROVED IF-PCA ALGORITHM

## ABSTRACT

Subject (e.g., cell or patient) clustering is an important problem in genetics and genomics. Influential features PCA (IF-PCA) is a recent idea for clustering, where we first select a small fraction of measured features and then cluster subjects (e.g., cells or patients) into different groups using the classical PCA clustering approach. A challenge the method faces is that we may have complex signal and noise structures across features or subjects or both, which may make the IF-PCA less effective. To deal with such a challenge, we propose a new approach, IF-PCA+, which combines IF-PCA with the recent idea of manifold fitting. The latter was shown to better support class separation. We compare our approach with the most popular subject clustering approaches, including but not limited to DESC, SC3 and Seurat, using 10 gene microarray data sets and 8 single-cell data sets. We show that with the new method, we have a significant improvement in feature selection accuracy, and that on average, our method outperforms several of the most competitive algorithms nowadays (including IF-PCA, DESC, Seurat) in terms of average rank and regret for clustering accuracy and adjusted Rand index. We also shed light on the insight underlying such improvements.

## 1 INTRODUCTION

Subject clustering, or high-dimensional clustering, uses measured features to group subjects such as patients or cells into multiple classes, and it is a problem of significant interest (e.g., [Eisen et al. (1998); Xu and Wunsch (2005); Jain (2010); Kiselev et al. (2019)]). Consider a setting where we have $n$ subjects from $K$ different classes (i.e., normal patients, diseased patients). For each $1 \leqslant i \leqslant n$, we have a class label $Y_i$ that takes values from $\{1, 2, \ldots, K\}$ and a $p$-dimensional measured feature $X_i$. The class labels are unknown, and the goal is to use $X_1, X_2, \ldots, X_n$ to estimate them.

In many applications (e.g., genomics [Eisen et al. (1998); Butler et al. (2018); Kiselev et al. (2019)], finance [Belloni and Chernozhukov (2011)], and astronomy [Takeuchi et al. (2024)]), an important observation is that, out of all $p$ features, only a small fraction of them are useful in deciding which subject belongs to which class. In such a setting, a reasonable approach to subject clustering is a two-step algorithm as follows. In the first step, we perform a feature selection and retain only a small fraction of the features which we consider as important. In the second step, let $\hat{S}$ be the indices of all retained features, and let $X_{\hat{S}}$ be the sub-matrix of $X$ with columns restricted to $\hat{S}$. We cluster all $n$ subjects into $K$ classes by applying the classical spectral clustering algorithm to $X_{\hat{S}}$.

The influential feature PCA (IF-PCA) proposed in [Jin and Wang (2016)] is an algorithm of this line. This algorithm has demonstrated empirical success in both microarray and single-cell data sets and has undergone rigorous theoretical evaluation (e.g., [Chen et al. (2023); Jin and Wang (2016)]). In particular, the approach has been shown to be optimal in the rare/weak signal model (e.g., [Chen et al. (2023)]).

With that being said, for complicated data sets such as microarray data and single-cell data, IF-PCA has room for improvements.

First, in previous studies on IF-PCA (e.g., [Jin and Wang (2016)]), the authors usually assumed that samples are independent, an assumption that does not always hold in single-cell data. Second, in the

feature selection step of IF-PCA, we test the significance of each feature one at a time, overlooking correlations among features. Third, IF-PCA is not designed for addressing high dropout noise. High dropout noise is frequently found in single-cell data sets, posing a well-known challenge for high-dimensional clustering. Last but not least, existing works on IF-PCA usually considered a model which we may call a *linear model*. In this model, we assume there are $K$ feature vectors $\mu_1, \mu_2, \ldots, \mu_K$ and $\mathbb{E}[X_i] = \mu_k$ if and only if sample $i$ belongs to class $k$. However, for single-cell data, such a model may not be appropriate, and the relationship between $Y_i$ and $X_i$ may be more complex and is likely to be significantly nonlinear.

To overcome these limitations, we propose IF-PCA+, an enhanced version of IF-PCA that integrates *manifold fitting* into the clustering process. By projecting data onto an estimated smooth manifold, manifold fitting can offer improvements in several directions.

First, manifold fitting recovers missing or distorted feature signals by addressing high dropout noise. In Section 4 Figure 1, we compared two approaches: one using sample-wise manifold fitting (as by [Yao et al. (2024)]), and the other using our new method, which combines sample-wise and feature-wise manifold fitting. Although both methods are effective at denoising and recovering true features, our approach, which utilizes correlations between samples and features, proves significantly more powerful. This suggests that manifold fitting is not only useful for removing high dropout noise, but also should be applied across both samples and features.

Second, manifold fitting enhances PCA clustering by improving class separation, i.e., (a) reducing intra-class distances and (b) increasing inter-class distances. For (a), the denoising effect of manifold fitting brings points within the same class closer together. For (b), in data sets with significant dropout noise (e.g., zero-inflated entries), points with many zeros are projected near the origin in spectral embedding. By filling these zero-inflated locations with meaningful signals from neighboring points, manifold fitting increases the separation between different classes.

The manifold fitting component of our algorithm, *diffusion-based manifold fitting (DMF)*, is an enhanced version of previous manifold fitting approaches. By utilizing diffusion maps, DMF improves robustness against high-dimensional noise during neighborhood selection and removes the need for additional transformations required by earlier methods, as demonstrated by empirical evidence in Section 2.1.

To integrate DMF efficiently with IF-PCA, we introduce two key modifications to the original IF-PCA algorithm. First, we remove the feature-wise normalization step. Second, we replace the fixed number of singular vectors used in the K-means clustering step with a more adaptive approach. These changes lead to a *modified IF-PCA* component that leverages the manifold structure of high-dimensional data more effectively.

Additionally, IF-PCA+ integrates both sample-wise and feature-wise manifold fitting, followed by the modified IF-PCA component, to address correlations across samples and features. To our knowledge, IF-PCA+ is the first algorithm to combine multiple manifold fitting steps for clustering tasks.

We benchmarked our approach against popular subject clustering methods, including but not limited to SC3 [Kiselev et al. (2017)], Seurat [Satija et al. (2015)] and DESC [Li et al. (2020)], using 10 gene microarray data sets and 8 single-cell data sets. For single-cell data sets, IF-PCA+ is the top-performing method in terms of average ranks and regrets for clustering accuracy and ranks second for the adjusted Rand index (ARI). Similarly, for microarray data sets, our method achieves the second-highest performance in terms of average ranks and regrets for clustering accuracy. Compared to IF-PCA, our method performs similarly on data sets where IF-PCA already achieves high accuracy (e.g., Deng, Darmanis, and Patel) and outperforms it on data sets where it faces challenges (e.g., Grun and Goolam).

In summary, by integrating both sample-wise and feature-wise manifold fitting and leveraging the enhanced DMF method with diffusion maps, IF-PCA+ is designed to handle data with correlations among features and samples while providing robust noise handling, especially in high-dimensional settings with complex noise structures such as zero inflation.

**Content**. In Section 2, we provide a detailed explanation of our main method. Section 3 presents the empirical results of IF-PCA+, comparing its performance against several recent methods using 8 single-cell data sets and 10 microarray data sets. Supporting simulation results are included in Section 4, and Section 5 concludes with a discussion.

## 2 METHODS

In the previous section, we discussed the limitations of IF-PCA and explained how the manifold fitting component, DMF, can be integrated with a modified IF-PCA to address these challenges. In this section, we will first introduce these two components separately and then present their combined application.

### 2.1 DMF: AN IMPROVED APPROACH FOR ROBUST MANIFOLD FITTING

Manifold fitting is a long-standing problem that has been addressed theoretically in the past several years by Fefferman et al. and Yao et al ([Fefferman et al. (2018; 0); Yao et al. (2023)]). Unlike popular manifold embedding methods, which project data onto a low-dimensional space, manifold fitting methods fit a low-dimensional manifold within the original space. One of the latest advancements, ysl23, proposed in [Yao et al. (2023)], ensures that the estimated manifold is smooth (twice differentiable) by using Euclidean nearest neighbors to estimate projection directions and then projecting points onto these directions. Building on ysl23, a clustering framework called Single-cell Analysis via Manifold Fitting (scAMF) was introduced in [Yao et al. (2024)]. The manifold fitting step of scAMF, which we referred to as Yao2, is a modification of ysl23 that consists of four steps: (a) correlation-based shared nearest neighbors (SNN) (b) value-to-rank transformation (c) estimation of the projection direction and (d) calculation of projection. Compared to ysl23, Yao2 tackles the curse of dimensionality with step (a), stabilizes variance through step (b), relaxes the smoothness condition in step (c), and simplifies calculations in step (d).

However, Yao2 still has places to improve. For one, its value-to-rank transformation is not optimal for dealing with heavy zero inflation. For another, its neighborhood selection relies solely on local information. To address these limitations, we propose *diffusion-based manifold fitting (DMF)* as an enhanced manifold fitting method utilizing diffusion maps. The DMF algorithm runs as follows.

Input: an $n$ by $p$ matrix $X$, a tuning parameter $knn$, the diffusion map dimension $K_{\text{diff}}$, and a diffusion bandwidth $\epsilon$. Output: a manifold fitted $n$ by $p$ matrix $M$.

- Diffusion neighborhood step. Use the data matrix $X$ with bandwidth $\epsilon$ to fit a $K_{\text{diff}}$ dimensional diffusion map $P$. Its projection into the diffusion space forms an $n$ by $K_{\text{diff}}$ coordinate map $C$, with each row $c_j$ in $C$ corresponding to row $x_j$ in $X$. For each $c_j$ with $1 \leqslant j \leqslant n$, find the closest $knn$ points using Euclidean distance in the diffusion space, and denote this set by $\mathcal{N}_{\text{diff}}(j)$. The diffusion-based shared nearest neighbor distance between point $x_j$ and point $x_\ell$ is $d_{\text{SNN}}(x_j, x_\ell) = |\mathcal{N}_{\text{diff}}(j) \cap \mathcal{N}_{\text{diff}}(\ell)|$. The $knn$ diffusion neighborhood of $x_j$ is defined by

$$\mathbb{B}(j) = \arg \max_{S \subset \mathcal{X}, |S| = knn} \sum_{x_\ell \in S} d_{\text{SNN}}(x_j, x_\ell).$$

- Estimate the projection direction component. For each $1 \leqslant j \leqslant n$, the projection direction $F(x_j)$ of $x_j$ is defined by

$$F(x_j) = \frac{1}{|\mathbb{B}(j)|} \sum_{x_\ell \in \mathbb{B}(j)} x_\ell$$

- Projection component. For each $1 \leqslant j \leqslant n$, the projection of $x_j$ onto the manifold, $m_j$, is computed as

$$m_j = \arg \max_{x_t} \rho(x_t),$$

where $x_t = x_i + t(F(x_i) - x_i)$ and $\rho(x_j) = \frac{1}{\sum_{x_\ell \in \mathbb{B}(j)} \|x_j - x_\ell\|_2^2}$. Construct a matrix $M = [m_1, \ldots, m_n]' \in \mathbb{R}^{n \times p}$.

**Differences between DMF and Yao2:** Differences between DMF and Yao2: Compared to Yao2, we have two major changes.

First, in Yao2, a correlation metric is used to measure the distance between two points. We replace the correlation metric with a diffusion metric. Unlike the correlation metric, which relies solely on local neighborhood information, the diffusion process and eigen-projection in diffusion maps capture both local and global relationships, making DMF more suitable for manifold data and more

robust against high-dimensional noise. For example, in the simulation setting described in Appendix Section F, where a 3D sphere is embedded into a 13111-dimensional space with Gaussian noise of variance 0.02, the diffusion-based SNN achieves an average neighborhood accuracy of 0.68, while the correlation-based SNN achieves only 0.34. This improvement in neighborhood accuracy is consistent across various noise levels and different 3D manifolds, including the torus, Swiss roll, and helix. Thus, diffusion-based SNN is a suitable choice for manifold data and is more robust against noise and the curse of dimensionality compared to correlation-based SNN in high-dimensional data.

Second, a value-to-rank transform is used in step (b) of Yao2 (see above). We find that after replacing the correlation metric with the diffusion-based metric, it is better to remove the value-to-rank transform. The reason is, diffusion maps naturally handle differences in scale and variance (through steps such as a normalized transition probability matrix) and stabilize variance (by choosing the leading eigenvectors while suppressing the influence of small-scale or noisy variations present in other eigenvectors). This makes the diffusion coordinates of each point carry specific meanings related to the intrinsic structure of the data and eliminates the need for a value-to-rank transformation.

We support these two main changes with some real data results below. As a benchmark, consider the procedure where we first apply Yao2, followed by IF-PCA. Alternatively, we first replace the correlation metric in Yao2 by the diffusion metric, and then remove the value-to-rank transform. We call this (tentatively) DMF. In Table 1, we compare the clustering accuracy of DMF with the benchmark, demonstrating a significant improvement using DMF. This finding supports the proposed changes. For details on parameter tuning, see Sections 2.3 and 3.1.

| Data sets | Camp1 | Camp2 | Darmanis | Deng | Goolam | Grun | Li | Patel |
|---|---|---|---|---|---|---|---|---|
| Benchmark (Yao2) | 0.751 | 0.514 | 0.743 | **0.871** | 0.822 | 0.513 | 0.918 | 0.914 |
| DMF | 0.799 | **0.64** | 0.768 | 0.813 | 0.806 | **0.92** | 0.921 | 0.925 |

Table 1: Improvements in clustering accuracy by using the two suggested changes. The improvement is especially significant for Camp2 and Grun.

To summarize, DMF is an improvement over Yao2 that is more robust against high-dimensional noise, accounts for manifold structure during neighborhood selection, and eliminates the need to handle tie-breaking for zero-inflated locations in the value-to-rank transformation.

## 2.2 Modified IF-PCA component

The IF-PCA implemented in IF-PCA+ have two modifications to the orthodox IF-PCA proposed in [Jin and Wang (2016)]. Firstly, we implemented a variant of orthodox IF-PCA, IF-PCA(X), that applies the PCA step directly to the input matrix instead of a normalized matrix, as proposed in [Chen et al. (2023)]. Since the input of the IF-PCA step in IF-PCA+ would be the manifold fitted result $\hat{M}$, a feature-wise normalization from the original IF-PCA would distort the shape of the estimated manifold, leading to loss of information and is thus unwanted.

Secondly, the PCA clustering step of the orthodox IF-PCA uses $(K-1)$ left singular vectors to run the k-means algorithm. However, after the projection of noisy data to a low-dimensional manifold, the $(K-1)$ left singular vectors may not be sufficient to capture useful signals of the manifold. This problem is especially significant for data sets with a complex data model but a small $K$ value (e.g., $K=2$). To remedy this problem, we use $K_0 = \max\{4, K\}$ top left singular vectors to run k-means. For more discussion on the choice of $K_0$, see Appendix Section B.

The *modified IF-PCA* runs as follows.

Input: an $n$ by $p$ matrix $X$, the number of classes $K$. Output: a predicted class label vector $\hat{Y} = (\hat{Y}_1, \hat{Y}_2, \ldots, \hat{Y}_n)$

- KS step. For each column (feature) $x(j)$ of $X$ (where $1 \leqslant j \leqslant p$), we compute a KS-score $\phi_n(x(j))$ using the formula

$$\phi_n(x(j)) = \sqrt{n} \sup_{t \in \mathbb{R}} \{\|F_{n,j}(t) - F_j(t)\|\},$$

where $F_{n,j}(t) = \frac{1}{n}\sum_{i=1}^{n}\mathbf{1}_{\{x_i(j)\leqslant t\}}$ is the empirical CDF of $x(j)$, and $F_j$ is the CDF for normal distribution $N(\bar{x}(j),\hat{\sigma}(j))$ with $\bar{x}(j) = \frac{1}{n}\sum_{i=1}^{n}x_i(j)$ and $\hat{\sigma}(j) = \left[\frac{1}{n-1}\sum_{i=1}^{n}\left(x_i(j)-\bar{x}(j)\right)^2\right]^{\frac{1}{2}}$.

- HCT step. Compute the empirical mean $\mu^*$ and standard deviation $\sigma^*$ of the KS-scores $\{\phi_n(x(1)),\ldots\phi_n(x(p))\}$. For each feature $j$, the normalized KS statistics is $\psi_j^* = \frac{\phi_n(x(j))-\mu^*}{\sigma^*}$, and the corresponding $p$-value is $\pi_j = 1 - F_j\left(\psi_j^*\right)$. Sort the $p$-values in ascending order, $\pi_{(1)} < \pi_{(2)} < \ldots < \pi_{(p)}$. The higher criticism threshold (HCT) is defined by $\hat{t}_{HC} = \pi_{(\hat{j})}$, where

$$\hat{j} = \text{argmax}_{\{j:\pi_{(j)}>\log p/p, j<p/2\}}\left\{\sqrt{p}\left(j/p-\pi_{(j)}\right)/\sqrt{\max\left\{\sqrt{n}\left(j/p-\pi_{(j)}\right),0\right\}+j/p}\right\}.$$

We retain feature $j$ if $\pi_j \leqslant \hat{t}_{HC}$ and remove it otherwise.

- PCA clustering step. Let $X^{IF}$ be the $n \times m$ sub-matrix of $X$ consisting only the retained features, where $m$ is the number of retained features. For any $1 \leqslant k \leqslant \min\{m,n\}$, let $\hat{\xi}_k^{IF}$ be the left singular vector of $X^{IF}$ corresponding to the $k$ th largest singular value. Define $K_0 = \max\{4,K\}$, and construct $\hat{\Xi}^{IF} = \left[\hat{\xi}_1^{IF},\ldots,\hat{\xi}_{K_0}^{IF}\right] \in \mathbb{R}^{n \times K_0}$. We cluster the $n$ subjects by applying $k$-means to the $n$ rows of $\hat{\Xi}^{IF}$, assuming there are $K$ clusters. Let $\hat{Y} = (\hat{Y}_1,\hat{Y}_2,\ldots,\hat{Y}_n)$ be the predicted class labels.

The Colon microarray data set (with $K = 2$) provides a good example of how the modified IF-PCA is more suitable for manifold fitting than the standard IF-PCA. Applying orthodox IF-PCA in IF-PCA+ results in a clustering accuracy of 0.516. In contrast, when IF-PCA+ is implemented with the modified IF-PCA step, the accuracy significantly improves to 0.838. For differences in performance across other data sets, we refer to Table 7 in the Appendix.

To summarize, for IF-PCA, we eliminate the feature-wise normalization step and replace the use of a fixed number of singular vectors in the k-means clustering step with a more adaptive approach. This adjustment improves the flexibility and performance of the method, allowing it to more effectively capture the manifold structure in high-dimensional data.

### 2.3 IF-PCA+

We now introduce our main method, *IF-PCA+*, which follows a pipeline consisting of data transformation, sample-wise manifold fitting, feature-wise manifold fitting, feature selection, and clustering.

Note that two input parameters in DMF, $K_{\text{diff}}$ and $\epsilon$, come from the diffusion map. In IF-PCA+, we first let $K_{\text{diff}} = \max\{4,K\}$ be the same with the $K_0$ in modified IF-PCA, then implement diffusion map using the 'pyDiffMap' python package, which automatically selects $\epsilon$ based on the bgh algorithm by Berry, Harlim and Giannakis [Berry et al. (2015)]. When bgh does not converge or the automatically selected $\epsilon$ is insufficient for the convergence of diffusion map (commonly in feature-wise manifold fitting), we set $\epsilon$ to be 5 times the variance of the Euclidean distance matrix computed between the rows of the input matrix. Thus, in practice, the combination of sample-wise and feature-wise DMF gives rise to two tuning parameters $knn_s$ and $knn_f$, respectively.

Depending on the signal strength (average number of samples per cluster), applying feature-wise manifold fitting may or may not be advisable, as the feature-wise signal strength in some data sets may be too low to reliably fit a manifold. For example, in the Goolam single-cell data set, $41\%$ of features have fewer than 20 nonzero entries. Therefore, a tuning parameter $n_0$ that determines when to apply feature-wise manifold fitting is necessary.

IF-PCA+ runs as follows. Input: an $n$ by $p$ matrix $X$, the number of clusters $K$, tuning parameters $n_0$, $knn_s$, and $knn_f$. Output: a predicted class label vector $\hat{Y} = (\hat{Y}_1,\hat{Y}_2,\ldots,\hat{Y}_n)$.

- Log-transformation. $X_{log} = \log(1+X)$.
- Sample-wise DMF. Let $K_{\text{diff}} = \max\{4,K\}$. Choose $\epsilon_s$ automatically based on the bgh algorithm. Apply DMF on $X_{log}$ with parameters $K_{\text{diff}}$, $\epsilon_s$ and $knn_s$ to obtain a new $n$ by $p$ denoised matrix $\hat{M}_s$.

- Feature-wise DMF. If $\frac{n}{K} > n_0$, apply DMF on transpose of $\hat{M}_s$ with parameters $K_{\text{diff}}$, $\epsilon_f$ (selected by bgh) and $knn_f$ to obtain an $n$ by $p$ denoised matrix $\hat{M}$.

- Modified IF-PCA. Apply modified IF-PCA to $\hat{M}$. Let $\hat{Y} = (\hat{Y}_1, \hat{Y}_2, \ldots, \hat{Y}_n)$ be the predicted class labels.

For data sets with meaningful negative values due to preprocessing (such as many microarray data sets), applying a log transformation is not advisable. For them, we skip the log transformation step and proceed with DMF steps to preserve data integrity.

**Tuning parameters** Selecting optimal values for $n_0$, $knn_f$ and $knn_s$ is difficult. For single-cell data sets, we set $n_0 = 100$, $knn_s = 15$, $knn_f = 50$ if $p < 10000$, and $knn_f = 100$ otherwise. The microarray data sets have smaller sample sizes and less severe dropout noise. For them, we set $n_0 = 50$, $knn_s = 10$ and $knn_f = 10$. For a discussion on the selection of tuning parameters in data sets without prior information, see Appendix D.

**Variants of IF-PCA+** There are multiple variants of IF-PCA+ one may consider:

(1) nolog-IFPCA+: IF-PCA+ without the log-transformation step;

(2) DMF-IFPCA: IF-PCA+ without the feature-wise manifold step;

(3) Yao2-IFPCA: IF-PCA+ without the feature-wise manifold step, with DMF replaced by Yao2;

(4) n0-IFPCA+: IF-PCA+ with both manifold fitting steps but without the threshold $n_0$;

(5) IF-PCA(W)+: IF-PCA+ with the original IF-PCA instead of the modified IF-PCA.

Each variant corresponds to a modification of a single step in IF-PCA+, allowing us to study how each individual step affects overall performance. The empirical comparison between IF-PCA+ and Yao2-IFPCA is presented in Section 3 Table 2. The emprical comparisons between IF-PCA+ and its other variants can be found in Appendix Section E.

From the comparisons, we observe that IF-PCA+ consistently delivers the best performance, demonstrating the effectiveness of our combined approach and highlighting the importance of each step. Thus, while IF-PCA+ may simplify to nolog-IFPCA+, DMF-IFPCA, or n0-IFPCA+ depending on data sets properties, the full IF-PCA+ is recommended for general use.

In summary, from the incorporation of diffusion maps to the development of modified IF-PCA and the integration of both sample-wise and feature-wise manifold fitting, each step in our algorithm is carefully designed to enhance its robustness, adaptability, and performance. This is evidenced by improved results in both simulations and empirical studies. Compared to the original IF-PCA, IF-PCA+ is more effective at handling nonlinear data, capturing correlations between features and samples, and providing robust noise management.

## 2.4 COMPUTATIONAL COMPLEXITY AND OTHER METHODS

In addition to IF-PCA and IF-PCA+, we also considered a range of state-of-the-art clustering methods including manifold learning techniques such as scAMF (Single-cell Analysis via Manifold Fitting) [Yao et al. (2024)], deep-learning algorithms such as IF-VAE [Chen et al. (2023)] and DESC [Li et al. (2020)], as well as established clustering methods like SC3 [Kiselev et al. (2017)] and Seurat [Satija et al. (2015)]. For a brief introduction to these methods, see Appendix Section A.

A asymptotic computational complexity analysis of these methods is provided in Appendix Section C. We find that, for large high-dimensional data sets, DMF has a lower computational complexity than Yao2, primarily due to the omission of the value-to-rank transformation step, which significantly increases the complexity in Yao2. When dealing with weak signal data, IF-PCA+ achieves a complexity of $O\left(n^2 p + p \log(p)\right)$, which is notably smaller than the manifold fitting method scAMF. For stronger signal data, IF-PCA+ incorporates feature-wise information, resulting in a complexity of $O\left(n^2 p + p^2 n\right)$.

To provide practical runtime insights, we report the running time on the Grun data set ($n = 1502$ and $p = 5547$). The methods DESC, IF-VAE, IF-PCA, and IF-PCA+ are implemented in Python, with the VAE steps of IF-VAE and the autoencoder step of DESC implemented through the Python 'keras' package. SC3 and Seurat are implemented in R using the SC3 package from Bioconductor and the

Seurat package, respectively. The manifold fitting method, scAMF, is implemented in MATLAB. The tuning parameters for these methods are the same as those described in Section 3.1 to ensure consistency.

In terms of runtime, Seurat (5 seconds) is the fastest, followed by IF-PCA (15.8 seconds), scAMF (24 seconds), IF-VAE (56 seconds) and DESC (80 seconds). The computational time of scAMF was reduced by a factor of 10 due to parallel processing. The main cost of IF-VAE arises from training the neural network. IF-PCA+ (1.9 minutes) and SC3 (2.6 minutes) are the slower methods. The most time-consuming part of IF-PCA+ is the feature-wise DMF, due to the larger $knn_f$ parameter. For SC3, the main cost is from computing the $n$ by $n$ similarity matrix.

The code to implement IF-PCA, IF-VAE, Seurat, SC3 can be found at https://github.com/ ZhengTracyKe/IFPCA. The code to implement scAMF can be found at https://github.com/zhigang-yao/scAMF. The code to implement IF-PCA+ is attached in Supplement.

## 3 RESULTS

Our study utilizes 8 scRNA data sets and 10 microarray data sets that were previously studied in Chen et al. (2023) for benchmarking purposes. These data sets cover embryonic, fetal, and adult tissues from both humans and mice. They have sample sizes between 100 and 2,000 and have been pre-processed using the code provided by the Hemberg Group under the column 'Scripts' of the link https://hemberg-lab.github.io/scRNA.seq.data sets. In all these data sets, true class labels are provided. However, we do not use the class labels in any of the clustering approaches. They are used only to evaluate the error rates. All data sets are freely available for download. For more detailed information on these data sets, see Section H of the Appendix.

### 3.1 COMPARISON OF CLUSTERING APPROACHES ON SINGLE-CELL DATA SETS

We first present results obtained from comparing IF-PCA+ with other methods in 8 single-cell data sets, see Table 2 and 3. Both clustering accuracies and ARI are used to rank methods for each data set, with a rank of 1 meaning the highest accuracy. We repeat each algorithm 10 times and report the average clustering accuracies. Similar to [Chen et al. (2023)], we also use regrets to evaluate the performance of methods. For each data set, the regret of a method is defined by $r = (e - e_{\min}) / (e_{\max} - e_{\min})$, where $e$ is the clustering error of this method, and $e_{\max}$ and $e_{\min}$ are the respective maximum and minimum clustering error among all the methods. The average regret also measures the overall performance of a method (the smaller, the better).

It is worth noting that log transformation and the use of top K left singular vectors can significantly improve the accuracy of some methods in single-cell data sets, such as IF-PCA (accuracy for Deng went from 0.588 to 0.828, and accuracy for Goolam went from 0.7 to 0.81). For a fair comparison, log transformation is applied on all data sets before the implementation of methods, top K left singular vectors are used for the clustering of IF-PCA, and PCA clustering steps of IF-PCA and IF-VAE are applied to the input matrix instead of a normalized matrix (as proposed in [Chen et al. (2023)]). Manifold fitting of IF-PCA+ and Yao2-IFPCA methods are implemented directly on log transformed data to avoid double log transformation.

When implementing scAMF and Yao2-IFPCA, we set $knn = 15$ in all data sets for consistency. Similarly, for IF-VAE, we fix $d = 25$ and use a mini-batch stochastic gradient descent with 50 batches, 100 epochs, and a learning rate of 0.0005 for all data sets. The parameter table of DESC can be found in Appendix Section A. For Seurat, we set $(m, N, k_0) = (1000, 50, 20)$ and choose different $\delta$ values for each data set to ensure that the number of clusters resulting from modularity optimization matched $K$. More details can be found in [Waltman and Van Eck (2013)]. The parameters for SC3 are set to $(x_0, d_0, k_0) = (10, 15, K)$. When implementing SC3 for the Patel data set, a gene filter needs to be dropped as it will remove all genes. Finally, we have the accuracy table, Table 2.

Results in table 2 show that:

(1) IF-PCA+ is the best performer, with the smallest average regret and average rank.

(2) IF-PCA remains competitive to modern algorithms despite its simplicity. It takes second place for average regrets and thrid place for average ranks.

| Accuracy | Seurat | SC3 | scAMF | DESC | IF-VAE | IFPCA | Yao2-IFPCA | IFPCA+ |
|---|---|---|---|---|---|---|---|---|
| Camp1 | 0.643 | 0.788 | 0.882 | 0.799 | 0.706 | **0.738** | 0.751 | **0.792** |
| Camp2 | 0.654 | 0.778 | 0.673 | 0.656 | 0.690 | 0.660 | 0.514 | 0.517 |
| Darmanis | 0.779 | 0.736 | 0.766 | 0.609 | 0.540 | 0.789 | 0.743 | 0.768 |
| Deng | 0.534 | 0.563 | 0.646 | 0.563 | 0.652 | 0.828 | 0.891 | 0.813 |
| Goolam | 0.629 | 0.758 | 0.823 | 0.629 | 0.492 | **0.721** | 0.822 | **0.806** |
| Grun | 0.993 | 0.500 | 0.523 | 0.968 | 0.750 | **0.673** | 0.601 | **0.966** |
| Li | 0.985 | 0.919 | 0.804 | 0.827 | 0.852 | 0.909 | 0.918 | 0.925 |
| Patel | 0.653 | 0.995 | 0.958 | 0.939 | 0.569 | 0.940 | 0.925 | 0.926 |
| Rank (mean) | 4.875 | 4.125 | **3.875** | 4.875 | 5.875 | **3.875** | 4.625 | **3.625** |
| Rank (sd) | 2.992 | 2.673 | 2.563 | 1.799 | 2.430 | 1.618 | 2.440 | 1.704 |
| Regret (mean) | 0.487 | 0.385 | 0.402 | 0.511 | 0.746 | **0.326** | 0.383 | **0.283** |
| Regret (sd) | 0.433 | 0.383 | 0.425 | 0.323 | 0.249 | 0.209 | 0.370 | 0.310 |

Table 2: Comparison of the clustering accuracies across the 8 single-cell RNA-seq data sets. IF-PCA+ is regarded as the best on average, with the smallest average rank and regret.

| ARI | Seurat | SC3 | scAMF | DESC | IF-VAE | IFPCA | Yao2-IFPCA | IFPCA+ |
|---|---|---|---|---|---|---|---|---|
| Camp1 | 0.519 | 0.763 | 0.801 | 0.729 | 0.639 | **0.629** | 0.651 | **0.685** |
| Camp2 | 0.425 | 0.594 | 0.484 | 0.483 | 0.464 | 0.490 | 0.335 | 0.231 |
| Darmanis | 0.719 | 0.700 | 0.667 | 0.526 | 0.428 | 0.703 | 0.670 | 0.675 |
| Deng | 0.427 | 0.541 | 0.561 | 0.426 | 0.431 | 0.848 | 0.886 | 0.843 |
| Goolam | 0.544 | 0.687 | 0.914 | 0.543 | 0.205 | **0.537** | 0.914 | **0.840** |
| Grun | 0.969 | -0.060 | -0.074 | 0.928 | 0.244 | **-0.096** | 0.104 | **0.853** |
| Li | 0.971 | 0.934 | 0.779 | 0.811 | 0.782 | 0.880 | 0.883 | 0.889 |
| Patel | 0.577 | 0.989 | 0.905 | 0.862 | 0.383 | 0.853 | 0.829 | 0.839 |
| Rank (mean) | 4.5 | **3** | 4 | 4.875 | 6.5 | 4.625 | 4.375 | **4.125** |
| Rank (sd) | 2.887 | 1.952 | 2.637 | 2.193 | 1.618 | 2.498 | 2.138 | 1.864 |
| Regret (mean) | 0.458 | **0.304** | 0.413 | 0.479 | 0.823 | 0.408 | 0.369 | **0.318** |
| Regret (sd) | 0.426 | 0.363 | 0.420 | 0.334 | 0.253 | 0.314 | 0.310 | 0.307 |

Table 3: Comparison of ARI values across the 8 single-cell RNA-seq data sets. On average, IF-PCA+ is a strong second in terms of rank and regret, showing significant improvements over IF-PCA in the Goolam and Grun data sets.

(3) In the Grun data set, where IF-PCA underperforms (and the Yao2-IFPCA method faces even greater challenges), IF-PCA+ achieves a high accuracy of 0.966.

(4) In data sets where IF-PCA performs relatively well, such as Deng and Darmanis, IF-PCA+ attains similar accuracies.

The adjusted Rand index (ARI) is another commonly used metric for clustering performance. In Table 3, we report the ARI values of different methods and recalculate the ranks and regrets.

From Table 3, we see that IF-PCA+ is ranked second in terms of ranks and regrets. IF-PCA, on the other hand, achieved similar performance to DESC and Seurat with a smaller average regret. Moreover, our new method brings significant improvements to the ARI in data sets where IF-PCA struggles, such as Goolam and Grun, while maintaining performance on par with IF-PCA in data sets where it performs relatively well.

In sum, the results from the two tables show that IF-PCA+ consistently outperforms other methods across a range of data sets, achieving the best average regret and rank for accuracy and a strong second place for ARI. It excels in challenging data sets like Goolam and Grun, where IF-PCA struggles, while also matching IF-PCA's performance in data sets where it already performs well, such as Deng. While IF-PCA remains competitive with modern algorithms, IF-PCA+ offers significant improvements in handling more difficult data sets, making it a more robust and versatile clustering method.

## 3.2 COMPARISON OF CLUSTERING APPROACHES ON MICROARRAY DATA SETS

We also benchmarked the clustering accuracy of IF-PCA+ on microarray data sets, see Table 4. In the table, IF-VAE uses a normalized data matrix $W$ (as in orthodox IF-PCA), while the IF-VAE(X)

| | Kmeans | SpecGem | IFVAE | IFVAE(X) | IFPCA | IFPCA+ |
|---|---|---|---|---|---|---|
| Brain | 0.667 | 0.857 | 0.500 | 0.500 | 0.738 | 0.738 |
| Breast Cancer | 0.562 | 0.562 | 0.565 | 0.572 | 0.594 | 0.600 |
| Colon Cancer | 0.548 | 0.516 | 0.597 | 0.597 | **0.597** | **0.838** |
| Leukemia | 0.972 | 0.708 | 0.722 | 0.833 | 0.931 | 0.967 |
| Lung Cancer(1) | 0.901 | 0.878 | 0.967 | 0.961 | 0.972 | 0.88 |
| Lung Cancer(2) | 0.783 | 0.567 | 0.783 | 0.783 | 0.783 | 0.727 |
| Lymphoma | 0.984 | 0.774 | 0.742 | 0.839 | 0.984 | 0.984 |
| Prostate Cancer | 0.578 | 0.578 | 0.588 | 0.598 | 0.618 | 0.588 |
| SRBCT | 0.556 | 0.492 | 0.524 | 0.635 | 0.556 | 0.460 |
| SuCancer | 0.523 | 0.511 | 0.672 | 0.672 | 0.667 | 0.638 |
| Rank (mean) | 3.55 | 5.2 | 3.5 | **2.85** | **2.15** | **3.15** |
| Rank (sd) | 3.505 | 5.62 | 3.3 | 2.585 | 2.115 | 3.215 |
| Regret (mean) | 0.558 | 0.868 | 0.605 | 0.421 | **0.188** | **0.357** |
| Regret(sd) | 0.426 | 0.312 | 0.421 | 0.366 | 0.253 | 0.407 |

Table 4: Comparison of clustering accuracy across the 10 microarray data sets. IF-PCA has the smallest average rank and average regret (boldface) and is regarded as the best on average. IF-PCA+ is ranked second.

uses unnormalized data matrix $X$. Our results indicate that, on average, IF-PCA performs the best on microarray data sets. The performance of IF-PCA+ is comparable to IF-VAE(X) and slightly surpasses that of other methods.

The reduced average regret between IF-PCA and IF-PCA+ indicates that IF-PCA, optimized for a rare/weak signal model under a Gaussian noise assumption, is sufficient for many microarray data sets. The additional manifold fitting steps may lead to overfitting on these data sets and could therefore be undesirable. It is worth noting that on the Colon data set (characterized by highly nonlinear patterns in its spectral embedding) where IF-PCA and other approaches struggled, IF-PCA+ achieves an accuracy of 0.838, demonstrating its effectiveness.

# 4 SIMULATIONS

In this section, we present simulation studies across four distinct settings to demonstrate how integrating manifold fitting enhances the feature selection process in IF-PCA+. For additional simulations demonstrating the improvement of DMF over Yao2 and IF-PCA+ over other methods, see Appendix Section F and G, respectively.

Inspired by the rare/weak model introduced in [Chen et al. (2023)], in all settings, we consider a noise matrix generated by $Z_{ij} \overset{iid}{\sim} N(0,1)$ and signal vector $\mu$ generated by

$$\mu(j) \overset{iid}{\sim} (1-\epsilon)\,\nu_0 + (\epsilon/2)\,\nu_\tau + (\epsilon/2)\,\nu_{-\tau}, \forall 1 \leqslant j \leqslant p,$$

where $v_a$ stands for point mass at a, and feature $j$ is an "influential feature" if $\mu(j) \neq 0$ and is a "noisy feature" otherwise. Specifically, the four distinct settings are:

1. Independent feature signals: $X_{ij} = Y_i \mu(j) + Z_{ij}$.

2. Correlated feature signals: $X_{ij} = Y_i U_j \mu(j) + Z_{ij}$, where $U_j \sim \text{Unif}(0.8, 1.2)$.

3. Correlated feature signals with dropout noise: $X_{ij} = (Y_i U_j \mu(j) + Z_{ij})\,\text{Ber}(q)$, with $q = 0.3$ and $U_j \sim \text{Unif}(0.8, 1.2)$.

4. Nonlinear correlated feature signals with dropout noise: $X_{ij} = (\log[1 + Y_i U_j \mu(j)] + Z_{ij})\,\text{Ber}(q)$, with $q = 0.3$ and $U_j \sim \text{Unif}(0.8, 1.2)$.

In all settings, we set $n = 500, p = 1000, K = 2, \epsilon = 0.5$ and $\tau = 0.25$. Class labels $Y_i$ are assigned such that the first half of samples are labeled "1" and the second half "-1".

To evaluate the enhancement of feature selection accuracy through manifold fitting in IF-PCA+, we compare the standard IF step with three manifold variants:

1. IF-s: Combines sample-wise DMF with IF.

2. IF-f: Combines feature-wise DMF with IF.

3. IF-sf: Integrates both sample-wise and feature-wise DMF with IF.

Feature selection accuracy is calculated by counting the number of influential features among the top 500 selected features, corresponding to the proportion $\epsilon = 0.5$. In all manifold steps, parameters are set as follows: $knn = 15, K_{\text{diff}} = 4$, and $\epsilon_{\text{diff}}$ is selected using the bgh algorithm. Each setup is repeated 10 times, yielding the accuracy results presented in Table 5.

|       | (1) Independent | (2) Correlated | (3) Corr-Dropout | (4) Log-Corr-Dropout |
|-------|-----------------|----------------|------------------|----------------------|
| IF    | 0.504           | 0.510          | 0.526            | 0.512                |
| IF-s  | 0.700           | 0.696          | 0.564            | 0.548                |
| IF-f  | 0.758           | 0.716          | 0.520            | 0.534                |
| IF-sf | 1.000           | 1.000          | 0.822            | 0.652                |

Table 5: Feature selection accuracy in the four simulation settings. The combined sample-wise and feature-wise approach, IF-sf, yields the best performance.

From Table 5, we observe that the combination of sample-wise and feature-wise DMF (IF-sf) consistently outperformed the original feature selection approach and single manifold approaches. IF-sf not only successfully identified all signal features in settings (1) and (2), which involved simple noise structure but a high noise level, but also demonstrated greater robustness to compound noise (Gaussian and dropout) and nonlinearity compared to other methods.

For a visual demonstration of the impact of manifold fitting on feature selection, see Figure 1.



Figure 1: Distributions of a simulated feature under setting (c). From left to right: (1) feature's true signal, (2) Observed feature (signal + noise), (3) Distribution after sample-wise Yao2, and (4) Distribution after sample-wise and feature-wise DMF.

From Figure 1, we see that although Yao2 is effective in recovering the signal of features from dropped-out noise, our new approach is much more powerful.

## 5 DISCUSSION

In this paper, we introduced IF-PCA+, an enhanced clustering algorithm designed to overcome the limitations of IF-PCA when applied to complex data sets such as microarray and single-cell data. By integrating both sample-wise and feature-wise manifold fitting, leveraging the enhanced DMF method with diffusion maps, IF-PCA+ demonstrates robust noise handling, especially in high-dimensional settings with challenging noise structures such as zero inflation.

Our two-step manifold fitting approach has more potentials. It can be combined with other feature selection steps and clustering methods, both linear and nonlinear. Our approach can also be applied to finance, network analysis, astronomy, and many other fields.

One limitation of our method is the need for three tuning parameters. How should one optimally combine two manifold fitting steps could be an interesting question to explore. Our method also implicitly assumes the existence of a low-dimensional smooth manifold, which may not hold true for other data sets, and can impact the generalizability of our approach.

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

## A BRIEF INTRODUCTION OF OTHER METHODS

IF-VAE is a deep-learning method introduced in [Chen et al. (2023)], that combines the feature selection step of IF-PCA with the variational auto-encoder (VAE). There are two versions of the IF-VAE proposed in [Chen et al. (2023)]. The first one applies the VAE to a normalized data matrix $W$ (same as orthodox IF-PCA), called IF-VAE, while the second one applies the VAE to the unnormalized data matrix $X$, called IF-VAE(X). The VAE component of the algorithm uses a traditional architecture in which both the encoder and decoder have one hidden layer; the encoder uses the ReLU activation, and the decoder uses the sigmoid activation. The only tuning parameter in this algorithm is the dimension of the latent space in the VAE, denoted by $d$.

Deep Embedding for Single-cell Clustering (DESC) is another autoencoder inspired clustering algorithm proposed in [Li et al. (2020)]. It uses a stacked autoencoder for data representation and integrates it with an iterative clustering neural network. When training DESC, we set parameters according to Table 6.

| | |
|---|---|
| # nodes 1st layer | 64 |
| # nodes 2nd layer | 32 |
| tolerance | 0.005 |
| # neighbors | 10 |
| batch size | 256 |
| louvain resolution | $[0.8, 1.0]$ |
| do tsne | TRUE |
| learning rate (TSNE) | 100 |

Table 6: Parameter table for DESC.

Seurat and SC3 are two recent algorithms that are especially popular for subject clustering with single-cell RNA-seq data. On a high level, both methods can be viewed as having a feature selection step and a clustering step. However, the clustering processes in both methods are complicated when compared to IF-PCA and IF-VAE with many tuning parameters.

Seurat was proposed in 2015 by Satija et al. [Satija et al. (2015)]. Its feature selection step filters genes based on their variability across cells, and its clustering step combines several methods, including PCA, the k-nearest neighbors algorithm, and modularity optimization. Due to the complexity, Seurat has four tuning parameters, $m, N, k_0, \delta$, where $m$ is the number of selected features in the feature selection step while $N, k_0, \delta$ pertain to the clustering step, corresponding to the PCA part, the k-nearest neighborhood algorithm part, and the modularity optimization part, respectively.

SC3 was first presented in 2017 by Kiselev et al. [Kiselev et al. (2017)]. Its feature selection step filters genes based on their expression levels, removing genes that are too rare or too ubiquitous. The

main idea of the clustering step in SC3 is to apply PCA many times (each for a different number of leading singular vectors) and use the results to construct a consensus matrix. We then cluster all subjects into $K$ groups using hierarchical clustering on the consensus matrix. There are three tuning parameters for SC3, one for the gene-filtering step, $x_0$, and two for the clustering step, $d_0$ and $k_0$, corresponding to the PCA part and the hierarchical clustering part, respectively.

## B  ADDITIONAL COMMENTS ON MODIFIED IF-PCA

Let $K_0$ denote the number of left singular vectors used in modified IF-PCA step. Here, we explain the rationale for setting $K_0 = \max\{4, K\}$.

Empirically, higher values of $K_0$ often yield better performance. For instance, in the Grun data set with only two clusters ($K = 2$), using the top 6 left singular vectors during IF-PCA+ achieves an accuracy of 0.879, while using 7 left singular vectors improved accuracy further to 0.886. In contrast, using only 4 left singular vectors resulted in a lower accuracy of 0.853 . However, increasing $K_0$ often comes with trade-offs, such as higher computational costs and a greater risk of overfitting, particularly in data sets with a small number of clusters or with complex noise structures.

Upon examining the scree plots of various single-cell data sets, including Camp1, Camp2, and Grun, we noticed a consistent pattern: the elbow point occured at 4-th singular values across these data sets. This observation strongly suggests that at least 4 dimensions are required to capture the core structure of the data. Additionally, to account for the loss of information during the fitting of a low-dimensional manifold, we adjusted $K - 1$ to $K$ for data sets with larger $K$ values to ensure robustness.

## C  ASYMPTOTIC COMPUTATIONAL COMPLEXITY OF METHODS

In this section, we discuss the asymptotic computational complexity of the methods used in our study. Given that the complexity of deep-learning algorithms varies significantly depending on their parameters, we focus our analysis on non-deep-learning algorithms.

**DMF:** For the manifold fitting method DMF, there are three main steps: the diffusion neighborhood step, the estimation of direction step, and the projection step. In the diffusion neighborhood step, the computational complexity is primarily determined by three components: the computation of the Gaussian kernel matrix during diffusion map construction, the eigen decomposition required to extract diffusion coordinates, and the computation of $k$-nearest neighbors in the resulting diffusion space. The Gaussian kernel computation has a complexity of $O\left(n^2 p\right)$. The eigen decomposition of the diffusion matrix contributes an additional complexity of $O\left(n^2 K_{\text{diff}}\right)$, where $K_{\text{diff}}$ is the number of retained diffusion dimensionality. Finally, finding the $k$-nearest neighbors in the diffusion space has a complexity of $O\left(n^2 p\right)$ in a naive implementation. Together, the diffusion neighborhood step has a total computational cost dominated by $O\left(n^2 p\right)$. Our implementation of the last two steps of DMF has a complexity of $O(np \cdot knn)$. With $knn$ significantly smaller than $p$, DMF has a computation complexity of $O(n^2 p)$.

**Yao2:** Similar to DMF, Yao2 has a complexity of $O(n^2 p)$ from finding shared nearest neighborhoods, a complexity of $O(np \log(p))$ from the value-to-rank transformation, and a complexity of $O(np \cdot knn)$ from both the estimation of direction step and the projection step. Together Yao2 has a complexity of $O(n^2 p + np \log(p))$.

**IF-PCA:** IF-PCA consists of three main steps: a feature selection step using Kolmogorov-Smirnov (KS) tests, an HCT step, and a PCA clustering step. The complexity of computing KS statistics for a single feature is $O(n \log(n))$, which results in a total complexity of $O(np \log(n))$ for all $p$ features. The HCT step involves sorting feature scores and has a complexity of $O(p \log(p))$. The PCA clustering step comprises two main components: singular value decomposition (SVD) and K-Means clustering, each with a complexity of $O(npK)$, where $K$ is the number of clusters. Therefore, the overall computational complexity of IF-PCA is $O(np \log(n) + p \log(p))$.

**IF-PCA+:** The IF-PCA+ algorithm integrates IF-PCA with both sample-wise and feature-wise DMF, resulting in a combined computational complexity of $O\left(n^2 p + p^2 n\right)$. However, for data sets with weak signals, feature-wise DMF is not implemented, reducing the complexity to $O\left(n^2 p + p \log(p)\right)$, which accounts for the sample-wise DMF and the PCA computation.

**scAMF:** With Yao2 being the dominant step of scAMF, scAMF also has a complexity of $O(n^2 p + np \log(p))$.

**SC3:** The five main steps of SC3 are: feature selection, pairwise distance calculations, PCA transformation, k-means clustering, and consensus clustering. Their respective computational complexities are $O(np), O(n^2 p), O(npK), O(n^2 p)$, and $O(n^2 \log(n))$, respectively. Overall, SC3 has a complexity of $O(n^2 p + n^2 \log(n))$. For high-dimensional data, SC3's complexity is dominated by $O(n^2 p)$.

**Seurat:** The clustering process in Seurat consists of the following steps: feature selection, data scaling, PCA, and SNN modularity optimization-based clustering (using the Louvain algorithm). The computational complexities of these steps are, respectively, $O(np + n \log n), O(np), O(npK)$, and $O(n^2)$ (or $O(k_0 n \log n)$ for sparse graphs, where $k_0$ is the parameter for KNN ). Thus, the dominating complexity is $O(npK + n^2)$.

**Summary:** For large high-dimensional data sets, DMF has a lower computational complexity than Yao2, primarily due to the omission of the value-to-rank transformation step, which significantly increases the complexity in Yao2. When dealing with weak signal data, IF-PCA+ achieves a complexity of $O\left(n^2 p + p \log(p)\right)$, which is notably smaller than the manifold fitting method scAMF. For stronger signal data, IF-PCA+ incorporates feature-wise information, resulting in a complexity of $O\left(n^2 p + p^2 n\right)$.

## D    TUNING PARAMETERS OF IF-PCA+

As discussed in Section 2.3, selecting optimal parameters is a difficult task. Without any prior knowledge about the data set, we need to first set a reasonable value for the number of clusters $K$, then proceed to select both $n_0$ and $knn$ parameters for DMF.

Selecting $K$: The selection of optimal $K$ in high-dimensional clustering is an open question. Without prior knowledge of the data set, a practical approach could involves initially estimating range of $K$ using methods like scree plots, then choosing a reasonable value using consensus matrices, as implemented in SC3.

A more theoretical approach may be to adapt the idea of st-GOF [Jiashun Jin and Wang (2023)] in network clustering to high-dimensional clustering. This approach entails modeling the data as $X = LM + Z = \Omega_k + Z$ (similar to the original work of IF-PCA), where $L \in R^{n,K}$ is the matrix where the $i$ th row is $e'_k$ (the $k$ th standard basis vector of $R^K, 1 \le k \le K$) if and only if Sample $i \in$ Class $k$, $M \in R^{K,p}$ is a matrix with each row corresponding to the mean signal of a cluster, and $Z$ representing the noise matrix. For each $k = 1, 2, 3, \ldots$, one may then estimate $\hat{\Omega}_k$. By defining $X_k = X - \hat{\Omega}_k$, one can define a test statistic using the idea of cycle-count, $T_k = \frac{\sum_{i_1,i_2,i_3,i_4} (\text{distinct}) X_k(i_1,i_2) X_k(i_2,i_3) X_k(i_3,i_4) X_k(i_4,i_1)}{C_k}$ with $C_k$ being a normalization constant. The goal is to demonstrate that $T_k$ converges to $N(0,1)$ when $k = K$ and diverges when $k \ne K$. This theoretical framework would provide a more rigorous method for determining the optimal number of clusters in high-dimensional data.

After determining a reasonable $K$, one can proceed to select parameters such as $n_0$, $knn_s$ and $knn_f$.

Selecting $knn_s$ and $knn_f$: The parameters $knn_s$ and $knn_f$ represent the number of nearest neighbors considered in the sample-wise and feature-wise DMF step. Since both parameters function similarly, we will focus on the selection of $knn_s$. Ideally, $knn_s$ should be less than the smallest cluster size to ensure meaningful local neighborhood structures. Without prior information on cluster sizes, a common heuristic is to set $knn_s$ close to $\sqrt{n}$. However, as computational demands increase with larger $knn_s$ values, it may be practical to threshold $knn_s$ at $\max\{\sqrt{n}, 100\}$. When working with multiple data sets of the same type, it is advisable to set $knn_s$ to the smallest $\sqrt{n}$ across all data sets. This strategy ensures consistency in parameter selection and facilitates comparative analyses.

Selecting $n_0$: The parameter $n_0$ in DMF determines whether a feature-wise DMF step is implemented. Serving as a lower bound, $n_0$ ensures that a sufficient number of samples are available to accurately recover the distribution of each feature. Without prior data set information, setting $n_0 = 50$ is advisable for noisy data. For data sets with particularly weak signals, increasing $n_0$ to 100 may yield more reliable results.

# E    Variants of IF-PCA+

| | scAMF | DESC | IFPCA | nolog-IFPCA+ | DMF-IFPCA | n0-IFPCA+ | IFPCA(W) | IFPCA+ |
|---|---|---|---|---|---|---|---|---|
| Camp1 | 0.881 | 0.799 | 0.738 | 0.789 | 0.795 | 0.792 | 0.791 | 0.794 |
| Camp2 | 0.673 | 0.656 | 0.66 | 0.547 | 0.634 | 0.517 | 0.537 | 0.522 |
| Darmanis | 0.7661 | 0.609 | 0.789 | 0.666 | 0.768 | 0.71 | 0.723 | 0.768 |
| Deng | 0.645 | 0.563 | 0.828 | 0.821 | 0.813 | 0.675 | 0.81 | 0.813 |
| Goolam | 0.823 | 0.629 | 0.721 | 0.83 | 0.806 | 0.677 | 0.806 | 0.806 |
| Grun | 0.5226 | 0.968 | 0.734 | 0.524 | 0.921 | 0.966 | 0.935 | 0.979 |
| Li | 0.8039 | 0.827 | 0.909 | 0.954 | 0.925 | 0.845 | 0.895 | 0.925 |
| Patel | 0.958 | 0.939 | 0.94 | 0.754 | 0.926 | 0.754 | 0.914 | 0.926 |
| Rank (mean) | 4 | 5.125 | 3.75 | 4.688 | 3.563 | 6.063 | 5.125 | 3.563 |
| Rank (sd) | 3.207 | 2.85 | 2.659 | 2.89 | 1.016 | 1.568 | 0.835 | 1.821 |
| Regret (mean) | 0.357 | 0.581 | 0.319 | 0.52 | 0.203 | 0.644 | 0.345 | 0.278 |
| Regret (sd) | 0.459 | 0.442 | 0.353 | 0.442 | 0.172 | 0.647 | 0.305 | 0.335 |

Table 7: Comparison of clustering errors for variants of IF-PCA+ across the 8 single-cell data sets.

The results in Table 7 show that IF-PCA+ shared the same average rank (0.563) with DMF-IFPCA, which outperforms all other variants of IF-PCA+ for single-cell data. This indicates that each step of our algorithm is meaningful.

Although it may be tempting to propose DMF-IFPCA as part of our final algorithm, the following table on microarray data, Table 8, reveals a performance gap. On the Lung Cancer(1) dat aset, IF-PCA+ shows a significant improvement over DMF-IFPCA, while performing similarly on other microarray data sets. This result suggests that DMF-IFPCA's gain in the Camp2 data set (the only data set where it outperformed IF-PCA+) may have been dataset-specific. Also, considering the significant accuracy gain observed in simulation setting of Section 4 and Appendix Section G, our proposal of double manifold fitting is well suited.

| | Brain | Breast | Colon | Leukemia | Lung(1) | Lung(2) | Lymphoma | Prostate | SRBCT | SuCancer |
|---|---|---|---|---|---|---|---|---|---|---|
| IFPCA+ | 0.738 | 0.616 | 0.838 | 0.967 | **0.934** | 0.727 | 0.984 | 0.588 | 0.460 | 0.638 |
| DMF-IFPCA | 0.738 | 0.616 | 0.838 | 0.967 | 0.872 | 0.724 | 0.984 | 0.588 | 0.460 | 0.638 |

Table 8: Comparison of DMF-IFPCA with IF-PCA+ across the 10 microarray data sets.

# F    Simulations for DMF

We are interested in studying the how neighborhoods of different metrics are affected under noise and the curse of dimensionality. The simulation procedure consists of four main steps:

- Generate ground truth neighborhood: Randomly generate 777 points of a 3D sphere in a 3D space. Use Euclidean distance in the original 3D space to define the ground truth neighborhood, as Euclidean distance is naturally meaningful in this space. For each point, find its top 15 nearest neighbors based on Euclidean distance in 3D space. This will act as the ground truth neighborhood.

- Embed data into 13111-Dimensional Space: The 777 points of 3D manifold give us a 777 by 3 matrix. Embed this matrix into a 13111-dimensional space by concatenating 13108 columns of entrywise Gaussian noise (with mean zero and variance $\sigma^2$) to the original 3D data. Call the new 777 by 13111 data matrix $A$. To simulate dropout effect, we further multiply $A$ entrywise with a Bernoulli random variable with probability $q = 0.3$, $X_{ij} = A_{ij}Ber(q)$, with $q = 0.3$. So matrix $X$ have a more complicated noise structure than $A$.

- Compute neighborhoods in 13111-dimensional space: Use the following four approaches to compute top 15 neighborhoods of each point:
  1. Euclidean Distance.

2. Correlation Distance.

3. Correlation-based Shared Nearest Neighbor (SNN) Neighborhood (Step (a) of Yao2).

4. Diffusion-based Shared Nearest Neighbor (SNN) Neighborhood (Step (a) of DMF).

- For each approach, compare the neighborhoods obtained in the 13111-dimensional space to the ground truth neighborhoods from the original 3D space. Compute the average accuracy over 5 runs: fraction of correctly identified neighbors in the high-dimensional space that match the ground truth neighbors in 3D space.

The size of the simulated matrix 777 by 13111 is the same with the Camp1 data set. For all $knn$-related parameters, we set $knn = 15$. For a diffusion map, we set $K_{\text{diff}} = 15$ and $\epsilon$ is selected automatically by bgh. From running the above simulation, we obtain Figure 2, which shows that our diffusion map based neighborhood method is significantly more robust against noise of various forms and curse of dimensionality. Specifically, we find that although an increase in Gaussian noise levels leads to a decrease in neighborhood accuracy (and consequently clustering accuracy), our new approach consistently outperforms others across all noise levels, demonstrating its robustness against high-dimensional noise. Moreover, Figure 2 illustrates that increasing the zero-inflation noise level from 0% to 30% does not alter the overall performance trend, furthur highlighting the robustness of our method to handle compound noise effectively. Similar results are observed when using other 3D manifolds such as helix, torus and Swiss roll.

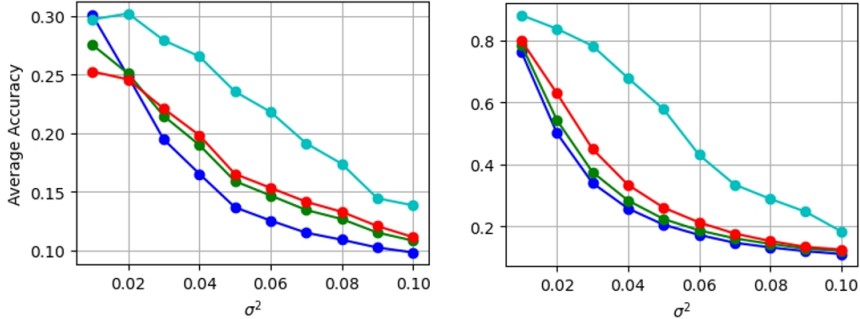

Figure 2: Left: Average accuracy of neighborhoods with no dropout noise. Right: Average accuracy of neighborhoods with a 30% dropout rate. Cyan: our proposed method. Blue: Euclidean distance. Green: correlation distance. Red: correlation-based SNN. Our proposed method significantly outperforms others across various noise levels, both with and without additional dropout noise.

## G   SIMULATION BENCHMARKS FOR IF-PCA+

The table below presents the clustering accuracy of various methods across the four simulation settings introduced in Section 4. Seurat demonstrates the highest clustering accuracy, with IF-PCA+ closely following. Notably, IF-PCA+ shows significant improvements over IF-PCA and scAMF, particularly in scenarios involving dropout noise and nonlinearity. In setting (d), IF-PCA+ achieves a clustering accuracy of 0.952, outperforming IF-PCA's 0.816 and scAMF's 0.502.

|         | (1) Independent | (2) Correlated | (3) Corr-Dropout | (4) Log-Corr-Dropout |
|---------|-----------------|----------------|------------------|----------------------|
| IFPCA+  | 1.0             | 1.0            | 0.980            | 0.952                |
| IFPCA   | 0.940           | 0.872          | 0.567            | 0.816                |
| scAMF   | 1.0             | 1.0            | 0.826            | 0.502                |
| SC3     | 1.0             | 1.0            | 0.976            | 0.947                |
| Seurat  | 1.0             | 1.0            | 0.988            | 0.974                |
| DESC    | 1.0             | 1.0            | 0.314            | 0.268                |
| IF-VAE  | 0.974           | 0.624          | 0.626            | 0.718                |

Table 9: Clustering accuracy of various methods across four simulation settings. IF-PCA+ demonstrates significant improvements over IF-PCA and scAMF.

# H DATA SETS

In our study, 8 scRNA data sets and 10 microarray data sets are used for benchmarking purposes. These data sets were previously studied in Chen et al. (2023).

The scRNA data sets cover embryonic, fetal, and adult tissues from both humans and mice. They have sample sizes between 100 and 2,000 and have been pre-processed using the code provided by the Hemberg Group under the column 'Scripts' of the link https://hemberg-lab.github.io/scRNA.seq.data sets. An additional filtering step has been applied so that genes with fractions of nonzero entries $< 5\%$ are filtered out. The resulting dimensions of these data sets are shown in Table 10.

The 10 microarray data sets, presented in Table 11, were also studied in [Jin and Wang 2016]. data sets 1, 3, 4, 7, 8, and 9 were analyzed and cleaned by Dettling (2004), while data sets 2, 6, and 10 were analyzed by Yousefi et al. (2010). Data set 10 was further cleaned by Jin and Wang (2016) using the same method as Dettling (2004). Data set 5 was obtained from Gordon et al. (2002). The 8 single-cell data sets can be download at https://data.mendeley.com/drafts/nv2x6kf5rd and the 10 microarray data sets can be downloaded at https://data.mendeley.com/data sets/cdsz2ddv3t.

| # | Data set | $K$ | $n$ | $p$ |
|---|----------|-----|-----|-----|
| 1 | Camp1 | 7 | 777 | 13,111 |
| 2 | Camp2 | 6 | 734 | 11,233 |
| 3 | Darmanis | 9 | 466 | 13,400 |
| 4 | Deng | 6 | 268 | 16,347 |
| 5 | Goolam | 5 | 124 | 21,199 |
| 6 | Grun | 2 | 1502 | 5,547 |
| 7 | Li | 9 | 561 | 25,369 |
| 8 | Patel | 5 | 430 | 5,948 |

Table 10: Single-cell RNA-seq data sets investigated in this paper. (n: number of cells; p: number of genes; K: number of cell types).

| # | Data name | Source | $K$ | $n$ | $p$ |
|---|-----------|--------|-----|-----|-----|
| 1 | Brain | Pomeroy (02) | 5 | 42 | 5,597 |
| 2 | Breast cancer | Wang et al. (05) | 2 | 276 | 22,215 |
| 3 | Colon cancer | Alon et al. (99) | 2 | 62 | 2,000 |
| 4 | Leukemia | Golub et al. (99) | 2 | 72 | 3,571 |
| 5 | Lung cancer (1) | Gordon et al. (02) | 2 | 181 | 12,533 |
| 6 | Lung cancer (2) | Bhattacharjee et al. (01) | 2 | 203 | 12,600 |
| 7 | Lymphoma | Alizadeh et al. (00) | 3 | 62 | 4,026 |
| 8 | Prostate cancer | Singh et al. (02) | 2 | 102 | 6,033 |
| 9 | SRBCT | Kahn (01) | 4 | 63 | 2,308 |
| 10 | Su cancer | Su et al. (01) | 2 | 174 | 7,909 |

Table 11: Microarray data sets investigated in this paper. (n: number of cells; p: number of genes; K: number of cell types).

