# OpenReview forum: "Subject Clustering by an Improved IF-PCA Algorithm"
_ICLR.cc/2025/Conference — Submitted to ICLR 2025_

### Official Review · Reviewer_DrVt · 2024-10-26

**Soundness:** 3
**Presentation:** 3
**Contribution:** 2
**Rating:** 6
**Confidence:** 3

**Summary:**

Clustering is an important problem in practice. In this paper, the authors consider subject clustering. They want to improve upon a previously proposed method, IF-PCA, which is to select features first, and then perform clustering of subjects using classical PCA clustering approach on the selected features. The authors argue that IF-PCA may not work well for problems with complex signal and noise structures. As a result, they proposed an improved version, IFPCA+ by combining manifold learning with IF-PCA. They show the performance using many real datasets.

**Strengths:**

The proposed IFPCA+ intends to overcome a few limitations of IF-PCA. Specifically, IF-PCA assumes samples are independent. Such an assumption doesn’t hold for problems such as single cell data. Furthermore, the feature selection step of IF-PCA doesn’t incorporate correlations among features, and IF-PCA only handles the data linearly. IFPCA+ aims to address these limitations through the use of manifold fitting with the diffusion-based manifold fitting (DMF) algorithm and its integration with IF-PCA. The authors have done comprehensive numerical comparisons with a number of alternative methods.

**Weaknesses:**

Although IFPCA+ is well motivated as an extension of IF-PCA, the method relies on the key step of manifold fitting. If such a low dimensional nonlinear structure exists, IFPCA+ has the great potential to handle data more effectively. However, it is not clear how to examine such an assumption for a practical problem. Furthermore, there are several tuning parameters to tune. How should one select them in practice without any knowledge about the potential number of clusters etc.? More discussion on the guideline will be helpful.

The authors mentioned a number of variants of the IFPCA+ on Page 6. It will be useful to give a clear recommendation on which one to use for a particular problem.

Please provide some discussion on the data size that the proposed method can handle in terms of n and p since in many genomic data, p can be in millions.

**Questions:**

1.	Single cell data often have zero-inflation. How does that affect the proposed clustering methods? Perhaps add some numerical simulation illustrates the effects of increasing proportions of zeros. The authors seem to suggest that the manifold fitting step helps to handle zero inflation. It is not clear to this reviewer why that is the case. More discussions on this would be needed.
2.	What are the numbers in Table 1? Please explain these numbers and how they were obtained for real data.
3.	For the PCA clustering step, the authors use max(4,K) top left singular vectors to run k-means. Why use 4 instead of a higher number?
4.	Does IF-PCA+ only work for non-negative data since log transformation was used for step 1 on Page 5? I saw a version of nolog-IFPCA+ mentioned later. It will be helpful to provide clear recommendations on which version to use for different applications.
5.	The writing needs to be improved for more clear presentation. For example, for the modified IF-PCA on Pages 4, 5, what does KS mean for the KS step? Are you referring to Kolmogorov-Smirnov test? On page 5, line 252, “tunning” should be “tuning”

---

> ### Author Response · Authors · 2024-11-28
>
> Many thanks for your detailed feedback on our manuscript.
>
> You brought up many good points:
>
> 1.	It is not clear how to examine if a low-dimensional manifold structure exists for a practical problem.
>
> 2.	How should one select parameters in practice without any knowledge about the potential number of clusters etc.?
>
> 3.	Give a clear recommendation on which variant of IF-PCA+ to use for a particular problem.
>
> 4.	Please provide some discussion on the data size that the proposed method can handle in terms of n and p since in many genomic data, p can be in millions.
>
> 5.	How does zero inflation affect the proposed clustering methods?
>
> 6.	What are the numbers in Table 1?
>
> 7.	Why use 4  in the max(4,K) adjustment for PCA clustering step instead of a higher number?
>
> 8.	Does IF-PCA+ only work for non-negative data?
>
> 9.	The writing needs to be improved for more clear presentation.
>
> Response to your main points:
>
> 1.	Thank you for your insightful comment regarding the key assumption of a low-dimensional nonlinear structure in the context of IF-PCA+. We agree that the effectiveness of IF-PCA+ hinges on the presence of such a structure. In practice, one may use elbow points of Scree Plots to check if the data have a low-dimensional structure. Alternatively, one may test the quality of manifold fitting using spectral embedding methods (e.g., diffusion maps), which visually or quantitatively reveal whether the data conforms to a low-dimensional structure. Last but not least, domain knowledge can also inform researchers whether a low-dimensional manifold structure is expected, as is often the case in genomic or single-cell data.
>
> 2.	Thank you for highlighting the practical challenges associated with selecting tuning parameters. To address this concern, we have added a section in the Appendix to provide practical (and theoretical) guidelines for parameter selection, particularly when there is no prior knowledge about the dataset.
>
> 3.	Thank you for your suggestion to provide clear recommendations on the use of different variants of IF-PCA+ for specific problems. In response, we have revised the corresponding subsection to clarify that the full IF-PCA+ is generally recommended for most scenarios due to its robust performance.
>
> 4.	Thank you for highlighting the importance of discussing the data sizes that the proposed method can handle. To address this concern, we have added details on the computational complexity of the proposed method in the revised manuscript. This discussion, combined with the provided runtime analysis, allows readers to estimate the scalability of the method for specific n and p values. While we do not explicitly state hardware requirements, the complexity analysis and the runtime data provide practical insights into the feasibility of applying our method to large genomic datasets.
>
> 5.	Thank you for raising the important question regarding zero inflation in single-cell data and its impact on the proposed clustering methods. Although not explicitly stated in the initial manuscript, our simulation for DMF in the Appendix demonstrates that an increase in noise levels leads to a decrease in neighborhood accuracy, ultimately resulting in worse clustering performance. Additionally, Figure 2 illustrates that although the increase in zero-inflation (dropout) noise level from 0% to 30% substantially reduces neighborhood accuracy, the general trend of performance does not change: our method consistently outperforms others. To address your concern, we have added this discussion to the Appendix in the revised manuscript, providing further insights on how zero inflation (and other noises) affects our methods.
>
> 6.	Thank you for pointing out the need for clarification regarding the numbers in Table 1. In the revised manuscript, we have added more detail to describe what these numbers represent and how methods were tuned to obtain these results.
>
> 7.	Thank you for your question regarding the use of \max(4, K) top left singular vectors for the PCA clustering step. In the revised manuscript, we have added a detailed discussion in the Appendix to explain this choice.
>
> 8.	Thank you for your comment regarding the applicability of IF-PCA+ to non-negative data. Yes IF-PCA+ does work with non-negative data by simply skipping the log-transformation step. In the revised manuscript, we have made this clear.
>
> 9.	Thank you for your feedback regarding the clarity of the writing and for pointing out specific areas for improvement. In the revised manuscript, we thoroughly reviewed and improved the writing for a clearer presentation.
>
> Thank you again for your time and valuable feedback!

---

### Official Review · Reviewer_g4yD · 2024-11-03

**Soundness:** 3
**Presentation:** 2
**Contribution:** 2
**Rating:** 6
**Confidence:** 3

**Summary:**

This paper presents IF-PCA+, an enhanced version of the influential feature principal component analysis (IF-PCA) algorithm, designed for high-dimensional subject clustering. IF-PCA+ incorporates diffusion-based manifold fitting (DMF) to improve feature selection, denoising, and class separation by leveraging both sample-wise and feature-wise manifold structures. The combined IF-PCA+ algorithm is designed to handle nonlinear relationships, correlations between features and samples, and robust noise handling. Experimental results show that IF-PCA+ outperforms several SOTA clustering methods on single-cell datasets and is competitive on microarray datasets.

**Strengths:**

1. The paper systematically identifies and addresses limitations in existing algorithms, progressively enhancing IF-PCA+ through the integration of DMF and an adaptive K-means clustering approach.
2. Experiments show that IF-PCA+ achieves the best average rank and regret across single-cell RNA-seq datasets, consistently outperforming other competitive methods.

**Weaknesses:**

1. The Methods section lacks a comprehensive description of prior work (e.g., ysl23, yao2, IF-PCA), making it difficult to follow. Some abbreviations (e.g., KS on line 209) are undefined, and citing experimental results (lines 174-185, 288-289) within the algorithm description seems unprofessional and unnecessary.
2. Comparisons use different languages (Python vs. R), leading to biased runtime results; theoretical complexity analysis would be fairer. Additionally, runtime results (lines 316-321) belong in the results section, ideally in a table for clarity.
3. Some abbreviations of comparative algorithms (Xs) are unclear and are only explained in the appendix. These explanations should be moved to the main text to improve readability and make the content flow more smoothly.
4. The parameters tuning part remains challenging.

**Questions:**

1. Table 4 shows a large regret gap between IF-PCA+ and IF-PCA, which needs further discussion.
2. Table 5’s IF-sf results reach 1.0 in settings A and B without a clear explanation, raising questions about robustness.

---

> ### Author Response · Authors · 2024-11-28
>
> Many thanks for your thoughtful feedback on our manuscript.
>
> You brought up five points:
>
> 1.	The Methods section lacks a comprehensive description of prior work (e.g., ysl23, yao2, IF-PCA), making it difficult to follow. Some abbreviations (e.g., KS on line 209) are undefined, and citing experimental results (lines 174-185, 288-289) within the algorithm description seems unprofessional and unnecessary.
>
> 2.	Comparisons use different languages (Python vs. R), leading to biased runtime results; theoretical complexity analysis would be fairer. Additionally, runtime results (lines 316-321) belong in the results section, ideally in a table for clarity.
>
> 3.	Some abbreviations of comparative algorithms (Xs) are unclear and are only explained in the appendix. These explanations should be moved to the main text to improve readability and make the content flow more smoothly.
>
> 4.	The parameters tuning part remains challenging.
>
> 5.	Table 4 shows a large regret gap between IF-PCA+ and IF-PCA, which needs further discussion.
> Table 5’s IF-sf results reach 1.0 in settings A and B without a clear explanation, raising questions about robustness.
>
> Response to your main points:
>
> 1.	We appreciate your comments about the comprehensiveness of prior work descriptions, the clarity of abbreviations, and the placement of experimental results.
> - Comprehensive Description of Prior Work: We acknowledge your concern about the descriptions of prior work, such as ysl23, Yao2, and IF-PCA. While we have not made significant changes to this section, we believe that the existing descriptions provide sufficient context to understand the basis for our proposed method. Furthermore, detailed discussions of these prior methods are available in their respective cited references, which we encourage readers to consult for additional clarity.
>
> -	Abbreviations:
> In response to your feedback, we have ensured that all abbreviations, including “KS” on line 209, are now properly defined when first introduced. We believe this change improves the overall readability and accessibility of the paper.
>
> -	Experimental Results in the Methods Section:
> While we understand your concern that including experimental results in the algorithm description may appear unconventional, we have retained these results as they directly support and validate specific methodological choices. For example, the cited results demonstrate why certain steps in the algorithm are necessary. We believe this integration provides readers with immediate insight into the rationale behind our approach, streamlining the presentation.
>
>
> 2. Thank you for pointing out the concern regarding comparisons using different programming languages, as well as the placement and format of runtime results.
> We agree that theoretical complexity analysis provides a fairer basis for comparison and have included a discussion in Section 2.4 with detailed analysis in the Appendix.
> Regarding the presentation of runtime results, we chose to retain the existing format in the Methods section as it is directly related to the description of the algorithms and their computational differences. That said, we appreciate the suggestion of presenting these results in a table and will consider this for future refinements.
>
> 3.	Thank you for pointing out the issue with unclear abbreviations of comparative algorithms (Xs) and suggesting that their explanations be moved to the main text.
>  To address this concern, we have removed most of the unclear abbreviations (Xs) from the manuscript. For the remaining abbreviations, we have added explanations directly in the main text where they are first introduced.
> 4. Thank you for your feedback regarding the challenges in the parameter tuning section.
> We acknowledge that parameter tuning can be complex, especially for methods involving multiple components like ours. To address this concern, we have included additional explanations and guidelines for parameter selection in the revised manuscript's Appendix, particularly in scenarios where there is no prior information about the dataset. This additional explanation will make the process clearer and more accessible to readers.
> 5. Thank you for pointing out the need for further discussion regarding Tables 4 and 5.
> To address your concerns, we have expanded the discussion for Table 4 to clarify the reasons behind the regret gap between IF-PCA+ and IF-PCA, which is likely due to overfitting, and provide an explanation of where IF-PCA+  sees an improvement over IF-PCA in microarray datasets.
> For Table 5, we have included more explanation on why an accuracy of 1.0 in settings A and B is impressive and added more interpretation to the result.
>
> Thank you again for your time and valuable feedback!

---

### Official Review · Reviewer_cSBb · 2024-11-03

**Soundness:** 3
**Presentation:** 2
**Contribution:** 3
**Rating:** 5
**Confidence:** 2

**Summary:**

The paper proposes an enhanced version of the Influential Features PCA (IF-PCA) method, referred to as IF-PCA+. The authors identify several limitations of the traditional IF-PCA, particularly its assumptions regarding sample independence, feature selection that overlooks correlations, and its ineffectiveness in handling high dropout noise commonly found in single-cell RNA sequencing (scRNA-seq) data. To address these challenges, the authors integrate a novel manifold fitting component, DMF (Data Manifold Fitting), with the modified IF-PCA. This combination aims to improve the robustness and accuracy of clustering in high-dimensional data. The paper reports that IF-PCA+ achieves competitive performance compared to modern clustering algorithms, demonstrating high accuracy in various datasets, including the Grun dataset and others where traditional IF-PCA struggles.

**Strengths:**

Originality: The paper presents a novel approach by integrating manifold fitting into the IF-PCA framework, resulting in the new method IF-PCA+. This proposal addresses several limitations of traditional IF-PCA, particularly in handling high-dimensional data with complex noise structures. The introduction of Diffusion-based Manifold Fitting (DMF) improves robustness against noise, leading to a significant advancement over clustering methodologies.

Significance: By improving clustering accuracy and feature selection in high-dimensional datasets, IF-PCA+ can facilitate better insights into biological processes and disease mechanisms. The findings could have broad applications in various domains, including cancer research and personalized medicine, making this work highly relevant and impactful.

**Weaknesses:**

The clarity of presentation of this paper needs to be improved. There are a lot of English mistakes, subject-verb disagreement, singular/plural noun errors, incorrect use of or missing articles (the/a/an), incorrect prepositions, verb form/conjugation errors, etc. Additionally, there are some typos in the mathematical formulas. These should be carefully addressed prior to the publication of this paper.

Examples:

ABSTRACT：
The phrase 'including IF-PCA, DESC, Seurat' is missing the conjunction 'and' before 'Seurat'.
Please define the acronym 'ARI' the first time it is used in the text.

1 INTRODUCTION
line 035: Change 'n subject' to 'n subjects'.
line 041: Change 'other feature are' to 'other features are'.
line 058: Change 'which poses' to 'posing'.
line 059: Change 'Last but not the least' to 'Last but not least'.
line 067: Change 'recover' to 'recovers' and 'by address' to 'by addressing'.
line 071: Change 'proves' to 'proves to be'.
line 073: Change 'feature' to 'features'.
line 078: Change 'neighborhoods' to 'neighboring'.
line 083: Add 'the' before 'empirical evidence'.
line 085: Change 'eliminated' to 'eliminating' and 'replaced' to 'replacing'.
line 087: Change 'resulting' to 'result' and 'utlize' to 'utilizes'.
line 092: Change 'with' to 'against'.
line 095: Change 'perform' to 'performs'.
line 096: Change 'Comparing to' to 'Compared to'.
line 097: Change 'perform' to 'performs' and 'achieve' to 'achieves'.
line 098: Change 'does not do well' to 'does not perform well'.
line 107: Change 'section' to 'Section' (to maintain consistency in capitalization).

2 METHODS
line 122: Change 'correlation based' to 'correlation-based'.
line 127: Change 'places' to 'room'.
line 133: Change 'an manifold' to 'a manifold'.
line 127: Correct the typo 'bandwith' to 'bandwidth'.
line 135: Change 'a n by Kdiff' to 'an n by Kdiff'.
line 155: Change 'Comparing to' to 'Compared to'
line 165: Change 'curse of dimensionality' to 'the curse of dimensionality' and 'when compared to' to 'compared to'.
line 169: Change 'stabilizing' to 'stabilize'.
line 177: Remove the redundant 'the' in 'the the benchmark'.
line 178: Change 'This support' to 'This supports'.
line 186: Change 'an improvement to' to 'an improvement over'.
lines 198, 200 and 202: Remove the redundant word 'many'.
line 213: Change 'x_i' in the subscript 'x_i \leq t' to 'x_j(i)', for notational consistency and clarity.
line 214: Change '\bar{x}(j)' to '\bar{x}_j' and '\hat{\sigma}(j)' to '\hat{\sigma}_j' , for notational consistency and clarity.
line 228: Change the subindex '-1' to 'K_0'.
line 233: Change 'for manifold fitting algorithms' to 'as a manifold fitting algorithm' or 'for manifold fitting'.
line 247: Add 'the' before 'diffusion map'.
line 252: Correct the typo 'tunning' to 'tuning'.
line 267: Change 'a n by p' to 'an n by p'.
line 313: Change 'matlab' to 'MATLAB' and 'same with' to 'the same as'.
lines 321 and 322: Change 'can be find' (which occurred twice) to 'can be found.

3 RESULTS
line 338: Change 'Table 2, and 3' to 'Tables 2 and 3'.
line 358: Add 'the' before 'gene filter'.

**Questions:**

In line 270, how to select the tuning parameters n_0, knn_f and knn_s?

---

> ### Author Response · Authors · 2024-11-28
>
> Many thanks for your positive feedback on the originality and significance of our paper and for carefully pointing out the grammatical errors in our manuscript.
>
> You brought up the point that:
>
> 1.	"The clarity of presentation of this paper needs to be improved. There are a lot of English mistakes … These should be carefully addressed prior to the publication of this paper."
>
> 2.	In line 270, how to select the tuning parameters $n_0$, $knn_f$ and $knn_s$?
>
> Response to your main point:
>
> 1.	We have addressed these issues by thoroughly revising the manuscript to enhance language clarity and grammatical accuracy. Additionally, we have included new simulations, expanded explanations, and further discussions to improve the flow and coherence of the paper.
>
> 2.	We have included additional explanations and guidelines for parameter selection in the revised manuscript's Appendix, particularly in scenarios where there is no prior information about the dataset. This additional explanation will make the process clearer and more accessible to readers.
>
> Thank you again for your time and valuable feedback!

---

### Official Review · Reviewer_QkBy · 2024-11-05

**Soundness:** 2
**Presentation:** 2
**Contribution:** 1
**Rating:** 3
**Confidence:** 5

**Summary:**

This paper proposes a novel strategy to cluster high-dimensional datasets, and applies it to number of important benchmark datasets, including both scRNA and micro-arrays.

**Strengths:**

- Clustering high-dimensional data is an important and unsolved problem
- Methods that work well on scRNA data are important
- Section 2.3 explains the IFPCA+ method well.
- The improvement of DMF over IF in Table 5 is impressive.

**Weaknesses:**

- I found the description of the method difficult to follow.  I do not need to know the history of developments of related methods.

- My summary of the work is that is it essentially a way of pre-processing the data to let IF-PCA run better (with slight modifications to the parameters of IF-PCA).  To the extent that it works, that is useful.  However, there is no theory suggesting when it would work.  The simulations are limited, in that I do not see comparison to the other methods in the simulation, nor do I know precisely which metrics are computed in the simulation.  It seems this work is more suitable for a venue like KDD.  To be appropriate for this venue, more theory would be appropriate.

- In the end, the method takes longer, improves a little in terms of some metrics of interest, but does not provide any more insight into the data.

**Questions:**

1. What is 'subject clustering', as opposed to 'clustering'?

2. Manifold fitting, both sample-wise and feature-wise, are new to me.  Is manifold fitting just another name for manifold learning? Please explain these concepts prior to using them.

3. What is a 'nonlinear dataset' (line 98 and elsewhere)

4. Some of the references are incorrect, please correct.

5. Please put all the background material in a background section.  The methods can then simply describe your method, highlighting the differences with previous work.

---

> ### Author Response · Authors · 2024-11-28
>
> Thanks for your valuable comments.
>
> You brought up points that:
>
> 1.	What is 'subject clustering', as opposed to 'clustering'?
>
> 2.	Manifold fitting, both sample-wise and feature-wise, are new to me. Is manifold fitting just another name for manifold learning?
> Please explain these concepts prior to using them.
>
> 3.	What is a 'nonlinear dataset' (line 98 and elsewhere)
>
> 4.	Some of the references are incorrect, please correct.
>
> 5.	Please put all the background material in a background section. The methods can then simply describe your method, highlighting the differences with previous work.
>
> Response to your main points:
>
> 1.	In the context of our work, "subject clustering" refers to grouping individual biological subjects (e.g., cells, patients, or tissue samples) based on their features or characteristics, as opposed to clustering features or other variables.
>
> 2.	Thank you for your question regarding manifold fitting and its distinction from manifold learning. While both manifold learning and manifold fitting focus on identifying the underlying geometric structures in data, traditional manifold learning methods primarily address the problem of "manifold embedding," which involves projecting data onto a lower-dimensional space. In contrast, manifold fitting focuses on projecting data onto a low-dimensional manifold that resides within the original high-dimensional space.
> We have added explanations of the concept in the methods section to enhance clarity.
>
> 3. Thank you for your question regarding the term "nonlinear dataset." A dataset is considered linear if its data matrix can be expressed as $X=L M+noise$ , where $L \in \mathbb{R}^{n, k}$ and $M \in \mathbb{R}^{k, p}$, representing a linear model. IF-PCA has been proven effective for such models. However, if this representation is not feasible, the dataset is classified as nonlinear, following a nonlinear model. We have revised the manuscript to avoid confusion.
>
> 4.	Thank you for pointing out the issue with incorrect references.
> We have carefully reviewed and corrected all references in the revised manuscript to ensure accuracy. This includes verifying citation details such as author names, titles, publication years, and journal information.
>
>
> 5.	Thank you for your feedback regarding the clarity of the method description and your suggestion to streamline the discussion by reducing background information.
> We acknowledge your concern and agree that excessive background information can distract from the main focus. However, manifold fitting is a relatively new concept that plays a key role in our approach. To avoid detracting from the primary discussion on the improvement to IF-PCA, we chose to briefly introduce manifold fitting in the Methods section rather than elaborating on it in the Introduction. This ensures that the focus remains on the main contributions of the paper while providing sufficient context for understanding the method.
> To address your concern, we have worked to simplify and condense the background content, ensuring it remains concise and essential without overcompromising the paper's readability or flow.
>
>
> Thank you again for your time and valuable feedback!

---

### Meta-Review · Area_Chair_SqCH · 2024-12-19

**Metareview:**

This paper addresses a method for clustering data points in a high-dimensional space.  Exemplary methods in this direction include IF-PCA and manifold fitting+clustering, which the proposed method is based on. The proposed method, referred to as "IF-PCA+" modifies IF-PCA, leveraging the idea of manifold fitting. Experiments on single cell RNA and micro-array data demonstrate the validity of the method.
The method seems to be sound. However, there are a few critical concerns that should be considered for future submissions. First of all, reviewers criticized that the description of the method is not easy to follow. It would be better to illustrate a big picture in a gentle way so that the readers can easily follow the rest of the paper. In general, the presentation should be improved since there are a few places where the description is not clear.  A few variants of IF-PCA+ are presented, but there is no sufficient descriptions for each of those, so that it is not clear which is a better choice depending on applications or situations. Therefore, the paper is not recommended for acceptance in its current form. I hope authors found the review comments informative and can improve their paper by addressing these carefully in future submissions.

**Additional Comments On Reviewer Discussion:**

The authors made efforts in responding but there was no change during the discussion period.

---

### Decision · Program_Chairs · 2025-01-22

Reject